# Diabetes Does Not Increase the Risk of Hospitalization Due to COVID-19 in Patients Aged 50 Years or Older in Primary Care—APHOSDIAB—COVID-19 Multicenter Study

**DOI:** 10.3390/jcm11082092

**Published:** 2022-04-08

**Authors:** Domingo Orozco-Beltrán, Juan Francisco Merino-Torres, Antonio Pérez, Ana M. Cebrián-Cuenca, Ignacio Párraga-Martínez, Luis Ávila-Lachica, Gemma Rojo-Martínez, Francisco J. Pomares-Gómez, Fernando Álvarez-Guisasola, Manuel Sánchez-Molla, Felix Gutiérrez, Francisco J. Ortega, Manel Mata-Cases, Enrique Carretero-Anibarro, Josep Maria Vilaseca, Jose A. Quesada

**Affiliations:** 1Health Center Cabo Huertas, Consejeria de Sanidad Univesal y Salud Pública, 03540 Alicante, Spain; dorozco@umh.es; 2Spanish Diabetes Society, 28002 Madrid, Spain; gemma.rojo.m@gmail.com; 3Clinical Medice Department, University Miguel Hernández, 03550 Alicante, Spain; gutierrez_fel@umh.es (F.G.); jquesada@umh.es (J.A.Q.); 4Endocrinology and Nutrition Service, University of Valencia, Hospital Universitari i Politècnic La Fe, 46026 Valencia, Spain; juan.merino@uv.es; 5Medicine Department, Autonoums University of Barcelona, 08193 Barcelona, Spain; 6Biomedical Research Network in Diabetes and Associated Metabolic Disorders (CIBERDEM), 20029 Madrid, Spain; 7Hospital Santa Creu i Sant Pau, Servicio Catalán de Salud, 08041 Barcelona, Spain; 8Primary Care and Prediabetes Group of the Spanish Diabetes Society, 30201 Cartagena, Spain; anicebrian@gmail.com; 9Health Center Cartagena Casco, Servicio Murciano de Salud, 30201 Cartagena, Spain; 10Primary Care Research Group, Biomedical Research Institute of Murcia (IMIB), 30120 Murcia, Spain; 11Spanish Society of Family and Community Medicine (semFyC), 28004 Madrid, Spain; iparraga@gmail.com (I.P.-M.); faguisasola@gmail.com (F.Á.-G.); 12Health Center Zone VIII, Servicio de Salud Castilla la Mancha, 02006 Albacete, Spain; 13Secretario GAPP-SED, Grupo DM-semFyC, 28004 Madrid, Spain; carlu91@gmail.com; 14Consultorio de Almáchar, UGC Vélez Norte, 29718 Malaga, Spain; 15Biomedical Research Institute (IBIMA), Endocrinology and Nutrition Clinical Management Unit, Malaga Regional University Hospital, 29010 Malaga, Spain; 16Diabetes Mellitus Plan of the Valencian Community, University Hospital San Juan de Alicante, 03550 Alicante, Spain; pomares_fra@gva.es; 17Health Center Ribera de Órbigo, Consejería de Salud Castilla León, 24280 León, Spain; 18Family Physician, Elche General University Hospital, 03203 Elche, Spain; manuel.sanchezm@umh.es; 19Internal Medicine, Elche General University Hospital, 03203 Elche, Spain; 20CIBER Infectious Diseases, 28029 Madrid, Spain; 21Health Center Campos-Lampreana, Conserjería de Salud Castilla y León, 49137 Zamora, Spain; fjortegarios@telefonica.net; 22Primary Care Center La Mina, Sant Adrià de Besòs, Servicio Catalán de Salud, 08930 Barcelona, Spain; manelmatacases@gmail.com; 23Group DAP-Cat, Research Support Unit, Jordi Gol University Institute for Primary Healthcare Research, CIBERDEM, 08036 Barcelona, Spain; 24Health Center José Gallego Arroba, Servicio Andauz de Salud, 14500 Córdoba, Spain; almudenayenrique@yahoo.es; 25Medicine Department, University of Barcelona, 08007 Barcelona, Spain; vilasesca@clinic.cat

**Keywords:** COVID-19, obesity and diabetes mellitus type 2, research, hospitalization, primary care

## Abstract

The purpose of this study was to identify clinical, analytical, and sociodemographic variables associated with the need for hospital admission in people over 50 years infected with SARS-CoV-2 and to assess whether diabetes mellitus conditions the risk of hospitalization. A multicenter case-control study analyzing electronic medical records in patients with COVID-19 from 1 March 2020 to 30 April 2021 was conducted. We included 790 patients: 295 cases admitted to the hospital and 495 controls. Under half (*n* = 386, 48.8%) were women, and 8.5% were active smokers. The main comorbidities were hypertension (50.5%), dyslipidemia, obesity, and diabetes (37.5%). Multivariable logistic regression showed that hospital admission was associated with age above 65 years (OR from 2.45 to 3.89, ascending with age group); male sex (OR 2.15, 95% CI 1.47–3.15), fever (OR 4.31, 95% CI 2.87–6.47), cough (OR 1.89, 95% CI 1.28–2.80), asthenia/malaise (OR 2.04, 95% CI 1.38–3.03), dyspnea (4.69, 95% CI 3.00–7.33), confusion (OR 8.87, 95% CI 1.68–46.78), and a history of hypertension (OR 1.61, 95% CI 1.08–2.41) or immunosuppression (OR 4.97, 95% CI 1.45–17.09). Diabetes was not associated with increased risk of hospital admission (OR 1.18, 95% CI 0.80–1.72; *p* = 0.38). Diabetes did not increase the risk of hospital admission in people over 50 years old, but advanced age, male sex, fever, cough, asthenia, dyspnea/confusion, and hypertension or immunosuppression did.

## 1. Introduction

Coronavirus type 2 is the cause of severe acute respiratory syndrome (SARS-CoV-2), better known as coronavirus disease 2019 (COVID-19), representing a major global health problem [1]. The infection presents an incubation period of around five days [2,3], after which the most frequent presenting symptoms are fever, dry cough, and fatigue, although other symptoms may also include productive cough, headache, hemoptysis, diarrhea, dyspnea, or lymphopenia [4,5,6].

Regarding the prognosis of the disease, 80% of cases are mild, 15% severe, and around 5% are critical; the case fatality rate is about 2% [7]. These figures are consistent with a technical document on the clinical management of COVID-19, wherein the Spanish Ministry of Health also estimates that approximately 80% of reported cases are mild [8]. In other countries, a hospitalization rate of 20 per 100,000 population between 1 January and 1 September has been described [9]. Applying this rate to the Spanish population would mean that around 6% of patients with COVID-19 and treated in ambulatory care would require hospitalization, leaving the vast majority of mild COVID-19 cases to be managed in primary care or on an outpatient basis. However, practitioners in these settings need to be able to identify the factors that increase the risk of severity and hospital admission in order to carry out an adequate assessment of the patient’s clinical prognosis and management and to assist in healthcare planning.

In that sense, some studies have suggested that diabetes mellitus is associated with a worse clinical prognosis [9], while others with large samples find no such relationship [10]. However, to our knowledge, there is no published information on prognostic variables predicting the need for hospitalization in patients with type 2 diabetes and COVID-19 who are treated in ambulatory care.

The aim of this study is to identify clinical, analytical, and sociodemographic variables associated with the need for hospital admission in people over 50 years of age who are infected with SARS-CoV-2 and followed in ambulatory care, and to specifically assess whether diabetes mellitus conditions the risk of hospitalization.

## 2. Materials and Methods

This retrospective case-control study was based on the analysis of variables included in the patients’ EMRs.

### 2.1. Selection Criteria

The study included all patients aged 50 and over diagnosed with COVID-19 based on laboratory tests and followed up in 41 participating primary care or outpatient endocrinology units, with home isolation. Patients who did not have laboratory confirmation of COVID-19 were excluded. Cases were defined as patients attended in ambulatory care and later admitted to the hospital due to COVID-19; controls were those who did not require admission.

### 2.2. Follow-Up Period

Patients were followed up retrospectively from 1 March 2020 until the cure date in cases of full ambulatory follow-up or the date of hospitalization in the admitted patients. The inclusion of patients ended on 30 April 2021.

### 2.3. Sample Size

The sample size was calculated to identify variables that increased the risk of hospitalization by 50% or more (odds ratio (OR) > 1.5), as described for different pathologies, including diabetes. Each study group required 173 patients, with an increase to account for missing data in an estimated 15% of EMRs. To ensure greater validity and representativeness in the control group, especially in variables with low prevalence, two controls were included for each case. Therefore, the estimated minimum sample size was 597 patients (199 cases and 398 controls).

### 2.4. Data Collection and Analysis

All data were collected retrospectively from the EMRs.

A descriptive analysis was performed by calculating frequencies for qualitative variables and the minimum, maximum, mean, and standard deviation (SD) for quantitative variables. The factors associated with hospital admission were analyzed using contingency tables, applying the chi-square test for qualitative variables, the Student’s t test for comparing means for quantitative variables, or nonparametric tests, as appropriate. To estimate the magnitude of the associations with hospital admission, logistic regression models were fit, using a simple adjustment for age and sex along with a multivariable adjustment. A stepwise variable selection procedure was performed based on the Akaike Information Criterion (AIC). Indicators of goodness-of-fit and predictive indicators such as the area under the receiver operating curve (AUC) are shown. Results are expressed as ORs with their 95% confidence intervals (CIs). Analyses were performed using SPSS (v.26) and R (v.3.6.1) software.

### 2.5. Ethical Aspects

The study complies with the principles of the Declaration of Helsinki for medical research involving humans and all relevant data protection laws. Data from EMRs were treated anonymously by assigning an individual patient identifier that did not allow linkage to the record number. The treatment, storage, and use of data complied with Organic Law 37/2018, of 5 December, on the Protection of Personal Data, as well as Regulation 2016/679 of the European Parliament and of the Council, of 27 April 2016, regarding the processing of personal data, as well as all applicable regulations and/or legislation. The Ethics Committee of San Juan University Hospital (Alicante) approved the study (code 20/025, dated 20 May 2020).

## 3. Results

A total of 790 patients throughout Spain were included by 61 researchers who had performed clinical care and follow-up in primary care centers or in outpatient endocrinology clinics (35%) (Appendix A). Of these, 495 who were not hospitalized during follow-up were assigned to the control group, and 295 who were admitted due to SARS-COV-2 were assigned to the case group.

Figure 1 describes the most frequent comorbidities present in the sample of patients studied. Hypertension was the most prevalent (50.5%), followed by dyslipidemia, obesity, and diabetes (37.5%). Just under half the patients (*n* = 386, 48.8%) were women, and 8.5% were active smokers. The distribution by age groups was as follows: <50–55 years, 18.1% (*n* = 143); 55–64 years, 31.9% (*n* = 252); 65–74 years, 24.6% (*n* = 194); 74–84 years, 18.2% (*n* = 144), and >84 years, 7.2% (*n* = 57). Table 1 describes the variables analyzed and their distribution between cases and controls, using a bivariable analysis.

Table 2a shows the results of the logistic regression adjusted for age and sex, and Table 2b shows a multivariate adjustment performed with a backward variable selection strategy, based on the AIC criterion, to arrive at an optimal model with all significant variables. The multivariable model used had a high explanatory power to assess the risk of hospital admission (AUC 0.860). The warning signs of a patient at risk of hospital admission were confusion, dyspnea, cough, and fever, while sore throat was associated with a lower probability of admission. Patient characteristics conferring a higher risk of hospitalization were age over 65 years and male sex, while the most relevant comorbidities were hypertension and immunosuppression. After adjusting for all other variables analyzed, neither diabetes nor obesity were associated with a higher risk of hospital admission in patients with COVID-19 followed in ambulatory care.

## 4. Discussion

This study included patients diagnosed with COVID-19 and followed-up in ambulatory care. Its main finding was that the risk of hospital admission was associated with the presence of certain symptoms (cough, fever, dyspnea, and/or confusion) along with male sex, age over 65 years, and comorbidities including hypertension or immunosuppression. However, the presence of diabetes was not independently associated with a higher risk of hospital admission.

Our sample had a rather high prevalence of several comorbidities (hypertension, dyslipidemia, diabetes, and obesity) (Table 1). This was due to its case-control design, where cases were defined by hospital admission. It is thus logical that the group of cases would be older with a higher prevalence of comorbidities than that in the general population. This was not a cross-sectional study, and its objective was not to describe the prevalence of diabetes but rather its possible association with the risk of admission.

One of the first indications regarding the relationship between diabetes and COVID-19 was the finding of a higher prevalence of diabetes among patients infected by COVID-19 [11,12]. Other studies have reported that the prevalence of diabetes is twice as high in people who died of COVID-19 (31%) compared with that in survivors (14%) [13]. In their meta-analysis, Puri et al. [14] identified 66 studies (39 in Asia and 27 in other regions) showing that the proportion of hypertension, diabetes, cardiovascular disease, and chronic kidney disease was significantly higher in patients with severe COVID-19 compared to that in patients with milder cases. However, these were prevalence studies, and the risk of hospitalization was not analyzed.

Other studies have focused on in-hospital mortality, observing that once adjusted for age, sex, degree of deprivation, ethnicity, and geographic region, the risk of in-hospital mortality doubled in people with type 2 diabetes (OR 2.03, 95% CI 1.97–2.09) and tripled in those with type 1 diabetes (OR 3.1, 95% CI 3.16–3.90) [12].

Regarding the risk factors for hospital admission observed internationally, Zhou et al. identified advanced age, the SOFA index (Sequential Organ Failure Assessment), and increased D-dimers as the most important [13]. In a study of 5416 adults in the USA [15], hospitalization rates were higher in patients with at least three comorbidities (aRR5.0, 95% CI 3.9–6.3), morbid obesity (aRR4.4, 95% CI 3.4–5.7), chronic kidney disease (aRR4.0, 95% CI 3.0–5.2), diabetes (aRR3.2, 95% CI 2.5–4.1), obesity (aRR2.9, 95% CI 2.3–3.5), hypertension (aRR2.8, 95% CI 2.3–3.4), and asthma (aRR1.4, 95% CI 1.1–1.7) after adjusting for age, sex, and race/ethnicity. Higher hospitalization rates were also seen in adults aged 65 years or older and in those aged from 45 to 64 years (vs. 18–44 years), in men (vs. women), and in non-Hispanic Black people and other races/ethnicities (versus non-Hispanic whites). Another study found higher rates of hospitalization in Black and Hispanic patients as well as at different poverty levels [16], and two meta-analyses have described increased risk of severe COVID-19 in patients with diabetes [17,18] and obesity [19]. In another study [20], patients with advanced age or comorbidities, including diabetes mellitus (in 28.3% of all patients), also had higher rates of hospitalization. Another meta-analysis presented similar results [21]. In Spain [22], studies in the first months of the pandemic reported that diabetes is associated with an increased risk of hospitalization and death, but the diagnosis of COVID-19 was a clinical suspicion, with no laboratory confirmation. Another study was done in patients with type 1 diabetes, confirmed COVID-19, and ambulatory follow-up showed age over 40 years as the main independent risk factor for hospital admission due to COVID-19, after adjusting for other variables such as HbA1c, sex, race, type of health insurance, and comorbidities [23]. Another study, this time in England [24], analyzed ambulatory EMRs to identify factors associated with COVID-19 mortality, finding that advanced age is the main factor, with risk increasing after age 60; a weaker association is also observed for diabetes. However, the interpretation of these results has been called into question [25].

With specific regard to patients with diabetes, poor glycemic control has been associated with an increased risk of complications in COVID-19 [26,27]. Hyperglycemia on admission also seems to be a risk factor for more complications and higher mortality [28]. As for antidiabetic treatment, a better prognosis has been observed in patients treated with metformin or sulfonyl ureas, while those treated with insulin are more likely to fare poorly [29]. In our subgroup of diabetic patients (*n* = 296), we observed that patients with poor glycemic control were at higher risk of hospitalization, but testing that association was not the objective of the study. Obtaining robust results about the impact of glycemic control on the evolution of the patient would require a larger sample and a specifically articulated research objective. For the same reason, antidiabetic treatments were not included in the analysis, since a larger sample of patients would be required to achieve valid results.

Many studies concluding that diabetes is associated with a worse prognosis in COVID-19 have a major limitation in that they either focus on the hospital setting (emergencies, admissions) or they do not differentiate between patients attended in ambulatory care versus those who are hospitalized or who present to the emergency department. Based on those data, it is impossible to identify the factors associated with the risk of admission in ambulatory patients, who represent approximately 85% of the population with COVID-19.

Another prevalent limitation in the literature is the lack of consideration for variables related to COVID-19 symptoms, though a meta-analysis of 12 studies and 3046 patients from the general population showed that fever, cough, fatigue, and dyspnea are associated with greater severity [30,31,32,33].

In our study, the bivariable analysis showed a significant association between the presence of diabetes and the risk of hospital admission, and the prevalence of diabetes was higher in patients who required hospital admission (46.8% vs. 31.9%; *p* = 0.001; Table 1). The same occurred with the presence of obesity (Table 1). However, after adjusting for the other included variables, including the clinical symptoms presented by the patient, the multivariable model could not confirm that diabetes and obesity were associated with an increased risk of hospital admission (Table 2). Rather, these factors were confounded by other variables (certain symptoms, advanced age, male sex, arterial hypertension), which more precisely determined the risk of admission. In fact, our multivariable model had a high explanatory capacity with an AUC of 0.86 (Table 2).

To our knowledge, this is the first study to identify prognostic variables related to the need for hospital admission in patients with COVID-19 followed in ambulatory services, considering both patient characteristics (age, sex, comorbidities) and the symptomatic presentation of COVID-19, which are the two criteria that are usually used in the follow-up of COVID-19 patients in clinical practice. We observed that the symptoms of the disease, as well as age and sex, were the predominant factors determining the risk of admission, outweighing some comorbidities such as diabetes or obesity. This result suggests that patients were not more likely to be admitted because of diabetes but because of advanced age, male sex, and presenting with fever, dyspnea, or confusion, among other symptoms. These data are not incompatible with the results of studies that relate diabetes to severity, since people with diabetes or obesity may present severe symptoms more frequently, but the symptoms and demographic characteristics confer a higher risk than the comorbidities themselves. All of this supports the applicability of these results to ambulatory practice in patients with COVID-19.

### Strengths and Limitations

The main strength of this study is the widely representative sample of patients, recruited by 61 researchers (family doctors and endocrinologists) throughout Spain, and the consideration of prognostic variables related to patient characteristics and COVID-19 symptoms (Appendix A). The inclusion of patients with and without diabetes, the confirmation of the diagnosis by laboratory tests, and the quality of the data collected by the attending physicians are also strengths of the study.

Limitations include the lack of data on some analytical parameters for assessing severity, which are routinely collected in the emergency department or during hospital admissions, but not in the ambulatory setting. For this reason, analytical prognostic variables such as neutrophils, lymphopenia, C-reactive protein, interleukin 6, serum ferritin, procalcitonin, or D-dimer were not included, as these tend to be requested only at the hospital level in patients presenting signs of severity in the emergency department or in those who are already admitted.

Finally, this study focused exclusively on assessing the risk of hospital admission. Mortality was not analyzed, since this would require a much larger sample size for the study design we applied. There were three out-of-hospital deaths that were not included in the analysis.

## 5. Conclusions

In patients over 50 years of age diagnosed with laboratory-confirmed COVID-19 and followed in ambulatory services, the risk of hospitalization was associated with symptoms such as cough, fever, confusion and dyspnea, underlying hypertension and immunosuppression, male sex, and advanced age. All these variables allowed the construction of a patient profile that would indicate a higher risk of admission: male, over 65 years of age, with high blood pressure or immunosuppression, who presented with cough, fever, dyspnea and/or confusion but not a sore throat. Finally, despite the relatively high prevalence of diabetes in included patients with COVID-19, diabetes was not independently associated with a higher risk of admission after adjusting for confounders. Although our findings suggest the potential role of these variables in developing hospitalization risk scores in ambulatory patients with COVID-19, regardless of the presence of diabetes, future studies designed to adequately evaluate their applicability in clinical practice are needed.

## Figures and Tables

**Figure 1 jcm-11-02092-f001:**
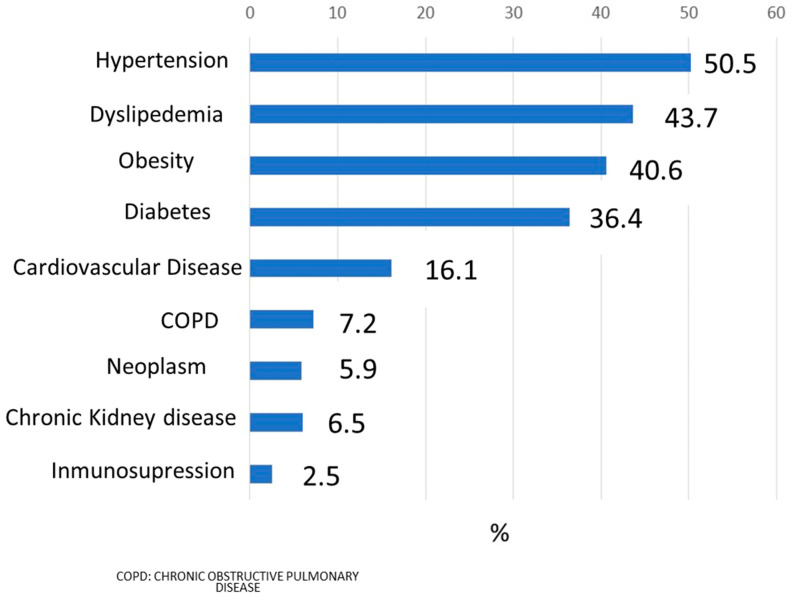
Most frequent comorbidities.

**Table 1 jcm-11-02092-t001:** Variables associated with hospital admission for COVID-19, bivariable analysis.

	Not Admitted (*n* = 495)		Admitted (*n* = 295)		*p* Value
	*n*	%	*n*	%	
Sex					
Female	277	56.0%	109	36.9%	<0.001
Male	218	44.0%	186	63.1%	
Age					
50–54 years	111	22.4%	32	10.8%	<0.001
55–64 years	174	35.2%	78	26.4%	
65–74 years	107	21.6%	87	29.5%	
75–84 years	74	14.9%	70	23.7%	
>84 years	29	5.9%	28	9.5%	
Body mass index					
<25 kg/m^2^	65	13.1%	44	14.9%	0.002
25–30 kg/m^2^	97	19.6%	71	24.1%	
>30 kg/m^2^	94	19.0%	78	26.4%	
Missing	239	48.3%	102	34.6%	
Active smoker					
No	334	67.5%	193	65.4%	0.81
Yes	42	8.5%	25	8.5%	
Missing	119	24.0%	77	26.1%	
O_2_ saturation					
Normal 90–100	204	41.2%	202	68.5%	<0.001
Low <90	1	0.2%	23	7.8%	
Missing	290	58.6%	70	23.7%	
Heart rate					
<60 bpm	6	1.2%	15	5.1%	<0.001
60–100 bpm	220	44.4%	184	62.4%	
>100 bpm	17	3.4%	23	7.8%	
Missing	252	50.9%	73	24.7%	
Systolic blood pressure					
<140 mmHg	217	43.8%	175	59.3%	<0.001
≥140 mmHg	75	15.2%	70	23.7%	
Missing	203	41.0%	50	16.9%	
Diastolic blood pressure					
<90 mmHg	260	52.5%	209	70.8%	<0.001
≥90 mmHg	32	6.5%	36	12.2%	
Missing	203	41.0%	50	16.9%	
Symptoms					
Fever	163	32.9%	222	75.3%	<0.001
Cough	174	35.2%	190	64.4%	<0.001
Asthenia/malaise	177	35.8%	193	65.4%	<0.001
Anorexia	26	5.3%	57	19.3%	<0.001
Myalgia	122	24.6%	103	34.9%	0.002
Dyspnea	47	9.5%	135	45.8%	<0.001
Productive cough	39	7.9%	42	14.2%	0.004
Sore throat	87	17.6%	29	9.8%	0.003
Diarrhea	59	11.9%	62	21.0%	0.001
Nausea/vomiting	31	6.3%	29	9.8%	0.067
Dizziness	17	3.4%	29	9.8%	<0.001
Headache	94	19.0%	66	22.4%	0.25
Shivering	37	7.5%	53	18.0%	<0.001
Loss of taste/smell	66	13.3%	33	11.2%	0.38
Chest tightness	18	3.6%	31	10.5%	<0.001
Confusion	2	0.4%	18	6.1%	<0.001
Comorbidities					
Hypertension	218	44.0%	191	64.7%	<0.001
Diabetes mellitus	158	31.9%	138	46.8%	<0.001
Dyslipidemia	210	42.4%	140	47.5%	0.17
Cardiovascular disease	68	13.7%	63	21.4%	0.005
Cancer	24	4.8%	23	7.8%	0.090
Chronic kidney disease	27	5.5%	25	8.5%	0.098
Immunosuppression	6	1.2%	14	4.7%	0.002
Gastrointestinal disease	28	5.7%	19	6.4%	0.65
COPD	25	5.1%	35	11.9%	<0.001
Asthma	28	5.7%	15	5.1%	0.73

bpm: beats per minute; COPD: chronic obstructive pulmonary disease.

**Table 2 jcm-11-02092-t002:** (a) Variables associated with hospital admission for COVID-19, multivariable analysis. Adjustment for age and sex. (b) Variables associated with hospital admission for COVID-19, multivariable analysis. Multivariable adjustment.

(**a**)				
		**Adjustment for Age and Sex**		
		**OR**	**95% CI**	** *p* **
Body mass index	<25 kg/m^2^	1		
	25–30 kg/m^2^	0.80	(0.48–1.35)	0.41
	>30 kg/m^2^	1.03	(0.62–1.72)	0.91
	Missing	0.52	(0.33–0.84)	0.008
Active smoker	No	1		
	Yes	1.07	(0.61–1.88)	0.80
	Missing	1.02	(0.71–1.46)	0.91
O_2_ saturation	Normal 90–100%	1		
	Low < 90%	25.05	(3.29–190.63)	0.002
	Missing	0.24	(0.17–0.34)	<0.001
Heart rate	<60 bpm	1		
	60–100 bpm	0.43	(0.15–1.18)	0.10
	>100 bpm	0.76	(0.23–2.52)	0.66
	Missing	0.15	(0.05–0.41)	<0.001
Systolic blood pressure	<140 mmHg	1		
	≥140 mmHg	0.94	(0.63–1.41)	0.77
	Missing	0.30	(0.21–0.45)	<0.001
Diastolic blood pressure	<90 mmHg	1		
	≥90 mmHg	1.36	(0.80–2.32)	0.25
	Missing	0.32	(0.22–0.47)	<0.001
Symptoms	Fever	6.68	(4.74–9.41)	<0.001
	Cough	3.48	(2.53–4.78)	<0.001
	Asthenia/malaise	3.58	(2.60–4.92)	<0.001
	Anorexia	4.35	(2.61–7.26)	<0.001
	Myalgia	1.85	(1.32–2.58)	<0.001
	Dyspnea	7.89	(5.31–11.7)	<0.001
	Productive cough	1.72	(1.06–2.80)	0.027
	Sore throat	0.51	(0.32–0.81)	0.004
	Diarrhea	2.01	(1.34–3.02)	0.001
	Nausea/vomiting	2.04	(1.17–3.58)	0.012
	Dizziness	3.18	(1.66–6.06)	<0.001
	Headache	1.43	(0.99–2.08)	0.060
	Shivering	2.80	(1.74–4.49)	<0.001
	Loss of taste/smell	1.00	(0.63–1.60)	>0.99
	Chest tightness	3.47	(1.86–6.48)	<0.001
	Confusion	10.35	(2.33–46.06)	0.002
Comorbidities	Hypertension	1.70	(1.23–2.35)	0.001
	Diabetes mellitus	1.43	(1.05–1.96)	0.024
	Dyslipidemia	0.99	(0.72–1.34)	0.93
	Cardiovascular disease	1.15	(0.77–1.72)	0.50
	Cancer	1.20	(0.64–2.22)	0.57
	Chronic kidney disease	1.10	(0.60–2.01)	0.76
	Immunosuppression	4.02	(1.46–11.07)	0.007
	Gastrointestinal disease	1.02	(0.55–1.91)	0.94
	COPD	1.61	(0.91–2.83)	0.10
	Asthma	0.73	(0.37–1.43)	0.36
(**b**)				
**Variables**		**Multivariable Adjustment**		
		**OR**	**95% CI**	** *p* **
Age	50–54 years	1		
	55–64 years	1.24	(0.69–2.22)	0.47
	65–74 years	2.45	(1.32–4.54)	0.005
	75–84 years	2.95	(1.52–5.73)	0.001
	>84 years	3.89	(1.7–8.9)	0.001
Sex	Female	1		
	Male	2.15	(1.47–3.15)	<0.001
Symptoms	Fever	4.31	(2.87–6.47)	0.000
	Cough	1.89	(1.28–2.80)	0.001
	Asthenia/malaise	2.04	(1.38–3.03)	<0.001
	Dyspnea	4.69	(3.00–7.33)	<0.001
	Sore throat	0.33	(0.18–0.58)	<0.001
	Confusion	8.87	(1.68–46.78)	0.010
Comorbidities	Hypertension	1.61	(1.08–2.41)	0.020
	Dyslipidemia	0.65	(0.44–0.96)	0.031
	Immunosuppression	4.97	(1.45–17.09)	0.011

bpm: beats per minute; COPD: chronic obstructive pulmonary disease. *n* = 790; N hospital admissions = 295. Likelihood ratio test, multivariable model (chi^2^ 338.3; *p* < 0.001); area under the receiver operating curve multivariable model = 0.860 (95% CI 0.835–0.886).

## Data Availability

All the data used for this analysis can be confirmed at any time.

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
