# Peer review of "Diabetes Does Not Increase the Risk of Hospitalization Due to COVID-19 in Patients Aged 50 Years or Older in Primary Care—APHOSDIAB—COVID-19 Multicenter Study"

_jcm, 2022, doi:10.3390/jcm11082092_

Round 1
Reviewer 1 Report
The study merely reports an exaggerated statistical Jaron, not a plausible occurrence.
1. Were the risks adjusted for age, gender and then recalculated for diabetes?
2. Too many tables have been presented directly from a statistical output, the inference is more important
3. Analysis of most clinical parameters as shown in Table 1 are likely to be significantly different- what did the authors gain by the analysis?
Author Response
Reviewer 1
Dear Reviewer,
We appreciate all the contributions you have made in your review and that have helped to improve the quality of this article. We have taken all your comments into account and made whatever changes were possible. Below we answer the questions raised.
- Were the risks adjusted for age, gender and then recalculated for diabetes?
Thank you for your comment. Table 2 shows the adjustment for age and sex and the optimal multivariate model. We have added ”Table 2 shows the results of the logistic regression adjusted for age and sex” on line 174, page 7.
In the adjustment for age and sex, it is observed that having diabetes presents some association with hospital admissions, with OR of 1.43 (1.05-1.96) with p-value 0.024, being weak evidence of this association, since the confidence interval is close to 1, and the p-value does not show great certainty that it is indeed different from 1.
A multivariate adjustment was performed with a backward variable selection strategy, based on the AIC criterion, to arrive at an optimal model with all significant variables. This optimal model does not include the variable diabetes, as the variables present in the model explain as much or more of the hospital admissions than diabetes. Therefore, having diabetes is not an independent predictor of hospital admissions.
The summary of the optimal model is shown in table 3.
- Too many tables have been presented directly from a statistical output, the inference is more important.
Thank you for your comment. The article has 3 tables:
Table 1 presents the incidence of hospital admissions in each explanatory variable. This table is important to understand the pattern of hospital admissions related to all variables, and to provide the reader with data on the sample sizes for each variable.
Table 2 shows the adjustment for age and sex, and the optimal multivariate model to explain the variables associated with hospital admissions. This table is necessary to meet the proposed objectives.
Both tables 1 and 2 contain inferential statistics, not descriptive statistics.
Table 3 is a summary of the optimal model in table 2. It may be useful for the reader to understand the fitted model more easily.
- Analysis of most clinical parameters as shown in Table 1 are likely to be significantly different- what did the authors gain by the analysis?
Thank you for your comment. The clinical parameters in table 1 regarding hospital admissions may be known, but this table provides the pattern of patients admitted at hospital level in our population, and we believe it is important to understand and discuss the role of diabetes in admissions, especially in relation to symptoms and other chronic diseases.

Reviewer 2 Report
This is a well-organized retrospective study in general. However, I still have some suggestions:
- In Figure1, hypertension has shown the most prevalent comorbidity. The authors should explain why they focused on the correlation test between COVID19 and diabetes but not hypertension.
- The title does not present the patient population is under the scope of aged > 50.
Author Response
Reviewer 2
Dear Reviewer,
We appreciate all the contributions you have made in your review and that have helped to improve the quality of this article. We have taken all your comments into account and made whatever changes were possible. Below we answer the questions raised.
- In Figure1, hypertension has shown the most prevalent comorbidity. The authors should explain why they focused on the correlation test between COVID19 and diabetes but not hypertension.
In multivariate analyses, both diabetes and hypertension are analysed. However, the novelty of this article is based on including not only the risk factors but also the symptomatology that the patient presents with COVID. It is therefore observed that, unlike other published studies that conclude the impact of diabetes on the risk of hospital admission, no such association is found. On the other hand, the association of hypertension with the risk of admission is confirmed, even taking into account the symptoms of COVID, so it is a better known result. Therefore, the most novel data that we think the study provides is that if the patient's COVID symptoms are taken into account, hypertension remains a risk factor for hospital admission, but diabetes loses this association.
2.The title does not present the patient population is under the scope of aged > 50.
Thank you for your comment. We propose this new title: Diabetes does not increase the risk of hospitalization for COVID-19 in patients aged 50 years or older in primary care. APHOSDIAB.COVID-19 multicenter study.

Round 2
Reviewer 1 Report
Few comments
Table 1, 2 and 3 mention all indications which are known indications of hospitalization (those with symptoms and geriatric age are likely to be hospitalized as per protocols)- This does not add anything to already known scientific literature
As can be seen in Table 2- Most of the analysis is redundant as can be seen with significant P values and CI intervals of MISSING values- No study can justify the same
The authors may remove the additional analysis and maybe just report their association with Diabetes as a Letter to Editor if permitted by Editorial Board
Author Response
Dear editor,
We appreciate all the contributions you have made in your review and that have helped to improve the quality of this article. We have taken all your comments into account and made whatever changes were possible. Modifications have been made using the track changes. Please, find the responses to your suggestions and questions below:
REVIEWER 1:
Table 1, 2 and 3 mention all indications which are known indications of hospitalization (those with symptoms and geriatric age are likely to be hospitalized as per protocols)- This does not add anything to already known scientific literatura.
The authors may remove the additional analysis and maybe just report their association with Diabetes as a Letter to Editor if permitted by Editorial Board.
Thank you for your comments. In multivariate analyses, both diabetes and hypertension are analysed. However, the novelty of this article is based on including not only the risk factors but also the symptomatology that the patient presents with COVID. It is therefore observed that, unlike other published studies that conclude the impact of diabetes on the risk of hospital admission, no such association is found. On the other hand, the association of hypertension with the risk of admission is confirmed, even taking into account the symptoms of COVID, so it is a better known result. Therefore, the most novel data that we think the study provides is that if the patient's COVID symptoms are taken into account, hypertension remains a risk factor for hospital admission, but diabetes loses this association.
As can be seen in Table 2- Most of the analysis is redundant as can be seen with significant P values and CI intervals of MISSING values- No study can justify the same
Thank you for your suggestion. To avoid redundancies in the analysis and achieve a better understanding of the reader, Table 2 has been disaggregated into Table 2a, which shows the variables associated with hospital admission for COVID-19 adjusted by age and sex, and Table 2b, which shows a multivariate adjustment with a selection strategy of backward variables, based on the AIC criterion, to arrive at an optimal model with all significant variables. ( page 5, line 153-156, and pages 6,7 and 8)
Table 3 has been removed.